# Re-evaluation of gestational age as a predictor for subsequent preterm birth

**Elizabeth Pereira[1], Gizachew Tessema[1], Mika Gissler[2,3], Annette K. Regan[1,4], Gavin Pereira**[1,5]*

**1** Curtin School of Population Health, Curtin University, Perth, WA, Australia, **2** Information Services Department, THL National Institute for Health and Welfare, Helsinki, Finland, **3** Department of Neurobiology, Care Sciences and Society, Karolinska Institute, Stockholm, Sweden, **4** School of Nursing and Health Professions, University of San Francisco, San Francisco, CA, United States of America, **5** Centre for Fertility and Health (CeFH), Norwegian Institute of Public Health, Oslo, Norway

* Gavin.f.pereira@curtin.edu.au

**Data Availability Statement:** The data underlying this study is not publicly available due to a data use agreement that restricts our ability to share the unit record data on ethical and legal grounds. The data are sensitive, and de-identified but potentially re-

## Abstract

### Background

To evaluate gestational age as a predictor of subsequent preterm birth.

### Materials and methods

This was a retrospective birth cohort study to evaluate gestational age as a predictor of subsequent preterm birth. Participants were mothers who gave birth to their first two children in Western Australia, 1980–2015 (N = 255,151 mothers). For each week of final gestational age of the first birth, we calculated relative risks (RR) and absolute risks (AR) of subsequent preterm birth defined as final gestational age before 28, 32, 34 and <37 weeks. Risks were unadjusted to preserve risk factor profiles at each week of gestation.

### Results

The relative risks of second birth before 28, 32, and 34 weeks' gestation were all approximately twenty times higher for mothers whose first birth had a gestational age of 22 to 30 weeks compared to those whose first birth was at 40 weeks' gestation. The absolute risks of second birth before 28, 32, and 34 weeks' gestation for these mothers had upper confidence limits that were all less than 16.74%. The absolute risk of second birth before 37 weeks was highest at 32.11% (95% CI: 30.27, 34.02) for mothers whose first birth was 22 to 30 weeks' gestation. For all gestational ages of the first child, the lowest quartile and median gestational age of the second birth were at least 36 weeks and at least 38 weeks, respectively. Sensitivity and positive predictive values were all below 35%.

### Conclusion

Relative risks of early subsequent birth increased markedly with decreasing gestational age of the first birth. However, absolute risks of clinically significant preterm birth (<28 weeks, <32 weeks, <34 weeks), sensitivity and positive predictive values remained low. Early gestational age is a strong risk factor but a poor predictor of subsequent preterm birth.

identifiable. Individual participant consent would be required to share these data but obtaining consent was waived by the Human Research Ethics Committee of the Department of Health, Western Australia. Rationale for this waiver included, but was not limited to, the potential for psychological harm among participants who experienced adverse perinatal outcomes and the infeasibility of obtaining retrospective consent for participants who gave birth to their children up to 40 years ago. Requests to access midwives notifications data can be submitted to the Data Linkage Branch of the Health Department of Western Australia, with enquiries directed here: DataServices@health.wa.gov.au. Information about the application process can be found at https://www.datalinkage-wa.org.au/apply/application-process/. Policies regarding data access can be found at https://www.datalinkage-wa.org.au/resources/policies/.

**Funding:** GP was supported with funding from the National Health and Medical Research Council (www.nhmrc.gov.au) Project and Investigator Grants #1099655 and #1173991 and the Research Council of Norway (www.forskningsradet.no) through its Centres of Excellence funding scheme #262700. The funders had no role in study design, data collection and analysis, decision to publish, or preparation of the manuscript.

**Competing interests:** The authors have declared that no competing interests exist.

# Introduction

It is widely accepted that previous preterm birth is a strong risk factor for preterm birth in later pregnancies [1–4]. Estimates from a large longitudinally linked cohort of approximately 200,000 births in Utah indicate that the risk of spontaneous preterm birth for women whose last birth was preterm was 4.7 to 5.4 times higher than women whose last birth was term [2]. Recurrence tended to occur at a similar gestational age to the previous preterm birth [1]. Moreover, there is increasing acknowledgment of patterns in morbidity across the continuum of gestational length [5].

Because previous preterm birth is a strong independent risk factor for subsequent preterm birth it can inform clinical management to prevent recurrence, such as screening for tocolytic treatment [6]. Family planning counselling requires a predictive or prognostic evidence base of absolute rather than relative risks of recurrent preterm birth because it is possible that previous preterm birth is explanatory but not predictive [7]. Relative risks, especially very low or high relative risks, can also be misinterpreted as absolute risk [8, 9]. Predicted gestational age based on the gestational age observed in the previous pregnancy can support the assessment of what to expect in the next pregnancy, can be used to initiate discussions to obtain patient perspectives when interpreting risk, and for obtaining consent to follow a management plan.

Population-representative summaries of gestational age based on the gestational age in previous pregnancy requires follow-up of enough mothers over a period sufficient to observe successive pregnancies with an adequate number of births at early gestations. The availability of linked individual-level perinatal data and population-wide coverage of births provides an opportunity to produce such summaries. The aim of this study was to estimate the absolute risk of preterm birth by gestational age of the previous pregnancy, and to estimate the predictive ability of the gestational age of the previous pregnancy.

# Materials and methods

## Study design and setting

This was a retrospective cohort study using probabilistically linked midwives notification data to estimate the absolute risk of preterm birth of the second birth based on the gestational length of the previous birth, for all notified singleton births in the state of Western Australia, 1980–2015.

## Participants and exclusions

Participants were women who gave birth to their first two singleton children in the study region, during the study period. From a starting population of 964,015 births, we sequentially excluded 26,730 multiple births; 5,187 births with missing final gestational age; 121 births with missing parity; 712 births with duplicate parity; and 1,872 births with gestational ages < 22 weeks or > 44 weeks to minimise influence of undetected late induced abortions and gestational age recording errors at the extremes of gestational age (Fig 1). Because this study focussed on recurrence, the final and largest exclusion (N = 419,091 births) was attributable to exclusion of births to women who only had one child during the study period and restriction to the first two births (parity 0 and 1). After these exclusions there were 255,151 mothers each with a parity 0 birth and a parity 1 birth (N = 510,302 total births).

## Variables and data sources

Unit records from the Midwives Notification System were obtained from the Western Australian Department of Health [10]. The Midwives Notification System is a legally mandated

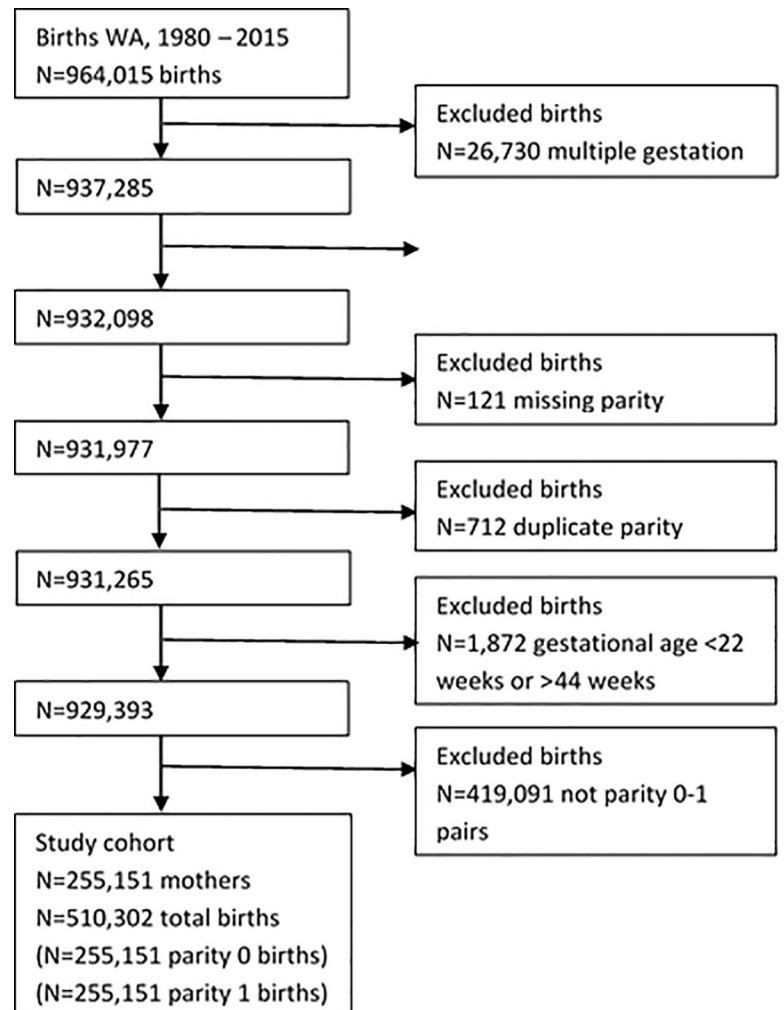

**Fig 1. Selection of the study cohort of first and second births to mothers in Western Australia (WA), 1980–2015.**

database of all births in the state attended by a midwife, nurse or doctor. The database includes information on maternal sociodemographics, maternal medical conditions, pregnancy, labor, delivery, the child and the child's separation. We used the clinical best estimate of final gestational age, based on ultrasound or last menstrual period. Gestational age was measured at the resolution of completed weeks of gestation. Midwives notifications are completed at the time of separation for all stillborn or live born neonates of $\geq$ 400 grams birth weight or $\geq$ 20 weeks' gestation. A de-identified unique number assigned to each mother and baby by the Western Australian Department of Health, Data Linkage Unit was used to identify births to the same mothers.

Ethical approval for this study was obtained from the Human Research Ethics Committee, Department of Health, Western Australia (#2016/51).

## Statistical analyses

As the aim of our study was to evaluate gestational age of the first child as a predictor for preterm birth of the second child we reported "absolute risk" of preterm birth and standard prediction metrics. These metrics provide information on the absolute degree to which the

gestational age of the first child indicates preterm birth of the second child. To evaluate gestational age of the first child as risk factor for preterm birth, we also report "relative risks". Relative risk provides information on the degree to which the gestational age of the first child indicates preterm birth of the second child compared to an alternative scenario (reference category). Relative risk in this study is the multiple by which risk of preterm birth of the second child is greater given a particular gestational age of the first child, relative to risk if the first child was born at full-term (40 weeks of gestation, reference category).

Gestational ages of the first child that were at the endpoints of the gestational age continuum were aggregated into categories (≤ 30, 31–32, 42–44 weeks) to preserve sample sizes. At each week (or category) of final gestation of the first child we calculated three measures of risk of preterm birth of the second child: (i) relative risk of birth of the second child at <28 weeks, <32 weeks, <34 weeks and <37 weeks compared to mothers whose first child been born at 40 weeks gestation; (ii) absolute risk of birth of the second child at <28 weeks, <32 weeks, <34 weeks and <37 weeks; and (iii) centiles (5th, 10th, 15th, 20th, 25th, 50th, 75th and 90th) of gestational age for the second child. Relative risk was calculated with log-binomial regression and 95% confidence intervals based on profile likelihoods for relative risks and Wilson score intervals for absolute risks [11, 12]. The definitions of preterm birth of the second child were selected to ensure that the cut-points were compatible with WHO classifications (*extremely preterm*, *very preterm*, *moderately preterm* and *preterm*) [13] with intentional overlap to maximise statistical power. Fewer centiles of the second child's gestational age above the median were selected because all pregnancies culminate at parturition at an upper bound of gestational age (set as 44 weeks in this study) resulting in lack of variability.

Because this was a prediction study, we preserved the distribution of risk factors at each gestational age of the first child. That is, we did not remove the effects of other risk factors by adjustment in regression models. Similarly, we did not restrict analyses to preterm birth of the first or second child by clinical presentation (spontaneous preterm birth versus medically indicated preterm birth). Because this was a large study with precise point estimates of relative and absolute risks, and to present conservative interpretation of results, we described findings based on interval estimates (bounds of 95% confidence intervals). Predictive ability was assessed using sensitivity, specificity, and positive and negative predictive values with 95% confidence intervals based on the Wilson score method. For this study positive predictive values are equivalent to absolute risk by gestational age of the first child and are reported separately.

We conducted a complete case analyses because of the very low proportion of missing values for gestational age (0.5%). Analyses were conducted with R v4.0.1 [14].

## Results

At first birth, mothers were more likely to be in the 25–29 years age group (34.7%), Caucasian (87.1%) and not smoke during pregnancy (13.6%) (Table 1). The index of relative socioeconomic disadvantage for the study population (mean of 1,007) was slightly higher than that of the national population (mean of 1,000) and less variable (standard deviation of 48 compared to 100). The prevalence of preterm birth before 37 weeks was 7.1% for the first child, 5.9% for the second child and 6.5% for both first and second births together.

### Relative risk of preterm birth of the second child by gestational age of the first child

Preterm births at second birth tended to recur at similar a gestational age to first birth (Table 2). The relative risk of second birth before 28, 32, and 34 weeks' gestation was approximately twenty times higher for mothers whose first birth had a gestational age of 22 to 30

**Table 1. Characteristics of the study population of mothers who gave birth to their first two singleton children between 1980 and 2015 in Western Australia (N = 255,151 mothers).**

|  | N (%) |
|---|---|
| All | 255,151 (100) |
| Maternal Age | |
| <20 years | 31,603 (12.4) |
| 20–24 years | 70,215 (27.5) |
| 25–29 years | 88,648 (34.7) |
| 30–34 years | 51,944 (20.4) |
| 35–39 years | 12,036 (4.7) |
| ≥40 years | 705 (0.3) |
| Ethnicity | |
| Indigenous [a] | 11,107 (4.4) |
| Caucasian | 222,149 (87.1) |
| Other | 21,895 (8.6) |
| Smoked during pregnancy [b] | |
| Yes | 17,287 (6.8) |
| No | 109,821 (43.0) |
| Missing | 128,043 (50.2) |
| Year of birth | |
| 1980–1984 | 33,892 (13.3) |
| 1985–1989 | 36,300 (14.2) |
| 1990–1994 | 37,514 (14.7) |
| 1995–1999 | 37,373 (14.6) |
| 2000–2004 | 37,477 (14.7) |
| 2005–2009 | 43,259 (17.0) |
| 2010–2015 | 29,336 (11.5) |
| Socioeconomic Index [c] | |
| Missing | 10,057 (3.9) |
| Mean (SD) [d] | 1,007 (48) |

a. Aboriginal or Torres Strait Islander.

b. Smoked during pregnancy. This variable was introduced in 1997.

c. Index of Relative Socioeconomic Disadvantage (IRSD).

d. SD: standard deviation. The national mean and standard deviation of the IRSD index are 1,000 and 100 respectively.

weeks compared to those whose first birth was at 40 weeks' gestation. For mothers whose first birth had a gestational age of 22 to 30 weeks, the risk of second birth before 37 weeks was approximately ten times higher than those whose first birth was at 40 weeks' gestation. The strength of associations decreased markedly with increasing gestational age of the first child. By 34 weeks' gestation of the first child, the interval estimates of the relative risk of preterm second birth (all classifications) were generally less than 10. By 37 weeks' gestation of the first child, the interval estimates of the relative risk of preterm second birth (all classifications) were less than 4. Interestingly, although the strength of associations decreased with increasing gestational age of the first child, relative risks remained statistically elevated (greater than 1) when the first child was born early term (37 and 38 weeks) and at 39 weeks, compared to those whose first birth was at 40 weeks' gestation, despite the small difference in gestational length (1–3 weeks).

**Table 2. Relative and absolute risk of preterm birth of the second child[a] by gestational age of the first child for the study population for mothers who gave birth to their first two singleton children between 1980 and 2015 in Western Australia (N = 255,151 mothers).**

| First Child | | Preterm Birth of Second Child | | | | Preterm Birth of Second Child | | | | Gestational Age of Second Child | | | | | | | |
|---|---|---|---|---|---|---|---|---|---|---|---|---|---|---|---|---|---|
| | | <28 weeks | <32 weeks | <34 weeks | <37 weeks | <28 weeks | <32 weeks | <34 weeks | <37 weeks | | | | | | | | |
| | N (%) | 960 (0.38) | 2,153 (0.84) | 3,678 (1.44) | 14,987 (5.87) | 960 (0.38) | 2,153 (0.84) | 3,678 (1.44) | 14,987 (5.87) | | Centile of Distribution | | | | | | |
| Gestational Age | | Relative Risk (95% Confidence Interval [b]) | | | | Absolute Risk (%, 95% Confidence Interval [c]) | | | | 5 | 10 | 15 | 20 | 25 | 50 | 75 | 90 |
| 22–30 weeks | 2,376 (0.93) | 19.00 (14.98, 23.99) | 21.49 (18.33, 25.17) | 19.46 (17.15, 22.06) | 10.05 (9.36, 10.79) | 4.63 (3.86, 5.55) | 10.35 (9.19, 11.64) | 15.24 (13.85, 16.74) | 32.11 (30.27, 34.02) | 28 | 31 | 33 | 35 | 36 | 38 | 39 | 40 |
| 31–32 weeks | 1,370 (0.54) | 9.29 (6.24, 13.34) | 11.97 (9.36, 15.10) | 13.52 (11.31, 16.05) | 10.03 (9.18, 10.93) | 2.26 (1.60, 3.19) | 5.77 (4.65, 7.13) | 10.58 (9.06, 12.32) | 32.04 (29.63, 34.56) | 31 | 33 | 34 | 35 | 36 | 38 | 39 | 40 |
| 33 weeks | 1,259 (0.49) | 4.89 (2.77, 7.97) | 8.57 (6.37, 11.30) | 10.86 (8.85, 13.18) | 8.98 (8.14, 9.86) | 1.19 (0.72, 1.96) | 4.13 (3.16, 5.38) | 8.50 (7.08, 10.17) | 28.67 (26.24, 31.23) | 32 | 34 | 35 | 36 | 36 | 38 | 39 | 40 |
| 34 weeks | 2,132 (0.84) | 5.58 (3.70, 8.11) | 7.89 (6.17, 9.95) | 8.75 (7.30, 10.41) | 7.89 (7.25, 8.56) | 1.36 (0.95, 1.95) | 3.80 (3.07, 4.70) | 6.85 (5.85, 8.00) | 25.19 (23.39, 27.07) | 33 | 34 | 35 | 36 | 36 | 38 | 39 | 40 |
| 35 weeks | 3,759 (1.47) | 3.49 (2.36, 5.01) | 5.25 (4.17, 6.54) | 6.08 (5.14, 7.16) | 6.53 (6.06, 7.03) | 0.85 (0.60, 1.20) | 2.53 (2.07, 3.08) | 4.76 (4.13, 5.49) | 20.86 (19.59, 22.18) | 34 | 35 | 36 | 36 | 37 | 38 | 39 | 40 |
| 36 weeks | 7,269 (2.85) | 2.99 (2.18, 4.03) | 3.54 (2.88, 4.33) | 4.53 (3.91, 5.24) | 5.25 (4.91, 5.60) | 0.73 (0.56, 0.95) | 1.71 (1.43, 2.03) | 3.55 (3.15, 4.00) | 16.76 (15.92, 17.63) | 34 | 36 | 36 | 37 | 37 | 38 | 39 | 40 |
| 37 weeks | 17,051 (6.68) | 1.88 (1.43, 2.44) | 2.48 (2.09, 2.95) | 2.82 (2.47, 3.21) | 3.56 (3.36, 3.77) | 0.46 (0.37, 0.57) | 1.20 (1.04, 1.37) | 2.21 (2.00, 2.44) | 11.37 (10.90, 11.86) | 35 | 36 | 37 | 37 | 37 | 38 | 39 | 40 |
| 38 weeks | 40,868 (16.02) | 1.44 (1.15, 1.79) | 1.67 (1.43, 1.94) | 1.80 (1.60, 2.02) | 2.08 (1.97, 2.20) | 0.35 (0.30, 0.41) | 0.80 (0.72, 0.89) | 1.41 (1.30, 1.53) | 6.65 (6.41, 6.89) | 36 | 37 | 37 | 38 | 38 | 38 | 39 | 40 |
| 39 weeks | 61,584 (24.14) | 1.17 (0.95, 1.44) | 1.25 (1.08, 1.45) | 1.30 (1.16, 1.46) | 1.38 (1.30, 1.46) | 0.28 (0.25, 0.33) | 0.60 (0.55, 0.67) | 1.02 (0.94, 1.10) | 4.40 (4.24, 4.57) | 37 | 37 | 38 | 38 | 38 | 39 | 40 | 40 |
| 40 weeks | 71,411 (27.99) | 1 | 1 | 1 | 1 | 0.24 (0.21, 0.28) | 0.48 (0.43, 0.54) | 0.78 (0.72, 0.85) | 3.19 (3.07, 3.33) | 37 | 38 | 38 | 38 | 38 | 39 | 40 | 41 |
| 41 weeks | 40,500 (15.87) | 1.11 (0.88, 1.41) | 1.04 (0.87, 1.24) | 0.93 (0.81, 1.07) | 0.83 (0.77, 0.89) | 0.27 (0.23, 0.33) | 0.50 (0.44, 0.57) | 0.73 (0.65, 0.82) | 2.65 (2.50, 2.81) | 37 | 38 | 38 | 38 | 39 | 40 | 40 | 41 |
| 42–44 weeks | 5,572 (2.18) | 0.74 (0.36, 1.32) | 0.93 (0.61, 1.37) | 1.03 (0.75, 1.38) | 0.93 (0.79, 1.08) | 0.18 (0.10, 0.33) | 0.45 (0.30, 0.66) | 0.81 (0.60, 1.08) | 2.96 (2.55, 3.44) | 37 | 38 | 38 | 38 | 39 | 40 | 41 | 41 |

a. Results reported to two decimal places, which is accurate when first child was not preterm (birth from 37 weeks). The results are accurate to 1 decimal place if the first child was preterm (birth before 37 weeks).

b. Profile likelihood confidence interval.

c. Wilson score confidence interval.

## Absolute risk of preterm birth of the second child by gestational age of the first child

The interval estimates for absolute risk of the second child birth before 28 weeks' gestation corresponding to all first child gestational ages were less than 5.55% (Table 2). The absolute risk of second child birth before 32 weeks' gestation ranged from 9.19% to 11.64% if the gestational

age of the first child was between 22 and 30 weeks, and less than 7.13% if the gestational age of the first child was more than 30 weeks. The interval estimates of absolute risk of second child birth before 34 weeks' gestation ranged from 13.85% to 16.74% if the gestational age of the first child was between 22 and 30 weeks, and was less than 12.32% if the gestational age of the first child was more than 30 weeks. In contrast, the absolute risk of second child birth before 37 weeks' gestation was elevated for all gestational ages of the first child less than 37 weeks, ranging from 10.90% to 34.56%.

### Distribution of gestational age of the second child by gestational age of the first child

The median gestational age of the second child was at least 38 weeks, and the lowest quartile (25th centile) was least 36 weeks (Table 2). The 10th centile of gestational age of the second child was less than 34 weeks if the gestational age of the first child was 22–30 weeks and 31–32 weeks, and was at least 34 weeks otherwise.

### Ability to predict preterm birth of the second child using preterm birth of the first child

Highest sensitivity for predicting preterm birth of the second child before 28, 32, 34 and 37 weeks' gestation was achieved using preterm birth of the first child before 37 weeks' gestation (Sensitivity, Table 3). However, for all categorisations of preterm birth, fewer than 35% of

**Table 3. Ability[a] of preterm birth[b] of the first child to predict preterm birth of the second child for the study population for mothers who gave birth to their first two singleton children between 1980 and 2015 in Western Australia (N = 255,151 mothers).**

| | | *Preterm Birth of Second Child* | | | | | | | |
|---|---|---|---|---|---|---|---|---|---|
| | | Sensitivity (%, 95% Confidence Interval [c]) | | | | Positive Predictive Value (%, 95% Confidence Interval [c]) | | | |
| | | <28 weeks | <32 weeks | <34 weeks | <37 weeks | <28 weeks | <32 weeks | <34 weeks | <37 weeks |
| *Preterm Birth of First Child* | <28 weeks | 8.02 (6.47, 9.91) | 6.73 (5.75, 7.87) | 5.30 (4.62, 6.07) | 2.66 (2.42, 2.93) | 6.09 (4.90, 7.54) | 11.46 (9.82, 13.34) | 15.42 (13.53, 17.51) | 31.54 (29.04, 34.15) |
| | <32 weeks | 12.92 (10.94, 15.19) | 13.05 (11.69, 14.54) | 11.85 (10.85, 12.94) | 6.35 (5.97, 6.75) | 4.17 (3.51, 4.95) | 9.44 (8.44, 10.55) | 14.65 (13.43, 15.97) | 31.99 (30.34, 33.69) |
| | <34 weeks | 16.25 (14.05, 18.72) | 17.51 (15.96, 19.17) | 16.69 (15.52, 17.93) | 10.43 (9.95, 10.93) | 3.12 (2.67, 3.64) | 7.53 (6.83, 8.30) | 12.27 (11.39, 13.21) | 31.23 (29.96, 32.53) |
| | <37 weeks | 28.13 (25.37, 31.05) | 31.44 (29.52, 33.44) | 32.54 (31.05, 34.08) | 27.37 (26.66, 28.09) | 1.49 (1.32, 1.67) | 3.73 (3.46, 4.01) | 6.59 (6.24, 6.96) | 22.58 (21.98, 23.20) |
| | | Specificity (%, 95% Confidence Interval [c]) | | | | Negative Predictive Value (%, 95% Confidence Interval [c]) | | | |
| | | <28 weeks | <32 weeks | <34 weeks | <37 weeks | <28 weeks | <32 weeks | <34 weeks | <37 weeks |
| | <28 weeks | 99.53 (99.51, 99.56) | 99.56 (99.53, 99.58) | 99.57 (99.55, 99.60) | 99.64 (99.61, 99.66) | 99.65 (99.63, 99.67) | 99.21 (99.17, 99.24) | 98.63 (98.58, 98.67) | 94.25 (94.16, 94.34) |
| | <32 weeks | 98.88 (98.84, 98.92) | 98.93 (98.89, 98.97) | 98.99 (98.95, 99.03) | 99.16 (99.12, 99.19) | 99.67 (99.65, 99.69) | 99.26 (99.22, 99.29) | 98.71 (98.67, 98.76) | 94.43 (94.34, 94.52) |
| | <34 weeks | 98.09 (98.04, 98.14) | 98.17 (98.12, 98.22) | 98.25 (98.20, 98.30) | 98.57 (98.52, 98.61) | 99.68 (99.66, 99.70) | 99.29 (99.26, 99.32) | 98.78 (98.73, 98.82) | 94.63 (94.54, 94.72) |
| | <37 weeks | 92.96 (92.86, 93.06) | 93.09 (92.99, 93.19) | 93.25 (93.15, 93.35) | 94.14 (94.05, 94.24) | 99.71 (99.69, 99.73) | 99.38 (99.34, 99.41) | 98.95 (98.91, 98.99) | 95.41 (95.32, 95.49) |

a. Sensitivity: Proportion of second preterm births for which the first birth was preterm; Positive Predictive Value: Proportion of first preterm births for which the second birth was preterm; Specificity: Proportion of second births that were not preterm for which the first birth was also not preterm; Negative Predictive Value: The proportion of first births that were not preterm for which the second birth was also not preterm.

b. Preterm: <28 weeks, <32 weeks, <34 weeks, <37 weeks.

c. Wilson score confidence interval.

mothers with a preterm born second child had a first child born preterm. Similarly, positive predictive values were highest when the preterm birth of the first child was before 28 weeks' gestation but fewer than 35% of mothers with a first child born preterm had a second child born preterm, for all categorisations of preterm birth (Positive Predictive Values, Table 3). Specificity and negative predictive values were all above 92%.

## Discussion

The absolute risks, sensitivity and positive predictive values observed for this Australian cohort imply that all categorisations of preterm birth of the first child were poor predictors for all categorisations of preterm birth of the second child. These results are reassuring for couples planning another pregnancy if their first child was preterm. The median gestational age of second births was 38 weeks (or more) irrespective of the gestational age of the first child. For the mothers whose first children were extremely preterm (<28 weeks) the absolute risk of clinically significant preterm birth (<34 weeks) was less than 18%.

Our results were derived from a large Australian cohort of singleton live births for which the prevalence of preterm birth in first pregnancy was 7.1% and 88% were Caucasian. The overall prevalence of preterm birth for this cohort (6.5%) is comparable to the Australian national estimate of 6.4% (95% CI: 6.3% to 6.6%) and the European estimate of 6.2% (95% CI: 5.8% to 6.7%) reported by the World Health Organisation [15]. In contrast, North America has greater sociodemographic diversity and a higher preterm birth prevalence of 10.6% (95% CI: 10.5% to 10.6%) [16]. Because patients' perspectives of risk of subsequent preterm birth should be based on individual clinical assessment and absolute risk, it would be prudent to use data from local cohorts, and therefore replication of our study in other settings is necessary. Until this study, there has been insufficient empirical research to inform mothers and clinicians of the range of gestational lengths to be expected in subsequent pregnancies.

Recurrence of birth at early gestations can be explained by biological, social and behavioural factors [17] that tend to be repeated between pregnancies. Although it is important to better elucidate these pathways, it is also essential to summarise empirically the range of expected gestational lengths regardless of pathways. Certain pathways, particularly those involving social and behavioural antecedents, will continue to be difficult to identify and it is plausible that a large proportion of variation in gestational length between pregnancies will remain unexplained. Therefore, summaries of absolute risk of preterm birth by week of gestation of the first child will remain useful for communication, particularly during counselling for postpartum family planning. Whether gestational length of the first child can be used for screening whole populations for subsequent preterm birth or other birth outcomes remains to be confirmed [18–21]. Our results indicate that gestational age of the first child is a poor predictor of preterm birth of the second child, but is a potential candidate for risk stratification if used with other information [16]. We note that preterm birth of the first child remains a strong risk factor for subsequent preterm birth. The high relative risks observed in our study might be explained by unobservable underlying processes that govern variation in gestational length that tend to repeat between pregnancies [17, 22].

As this was a large population based study, the estimates presented are precise and we had extensive information from midwives' notifications on the pregnancies. However, we did not have information on treatments that may have been applied to prevent preterm birth if the first child was born preterm. This would extend gestational length of the second child and therefore decrease absolute risks of preterm birth. Although prophylactic treatment would have become more common over time [23], allowing treatment to influence the absolute risk estimates can be interpreted as favourable because resultant estimates would reflect a more

accurate assessment of risk to be expected in subsequent pregnancy. The long duration of data collection enabled observation of larger numbers of births at early gestations (closer to viability) than if the duration of the study was shorter, but the longer study period is also accompanied by temporal changes. These temporal changes include the increasing use of more accurate means for estimating gestational age (ultrasound dating), changes in antenatal care, and changes in behavioural risk factors. Temporal changes during the study period would potentially inflate the confidence intervals reported in our study (loss of precision). Finally, we note that the main limitation of this study was the lack of validation of our results with an independent study population. Further studies in independent populations are needed to replicate our results. Such studies would require a large cohort of first births to observe births at early gestational ages and sufficient follow-up to observe second births.

## Conclusion

Gestational age is strongly associated with subsequent preterm birth, which supports findings of previous studies, but is a poor predictor of subsequent preterm birth. Communicating the absolute risks presented in this study has potential to minimise unintentional misinformation provided during postpartum counselling of mothers who have experienced a preterm birth and want to know their risk of experiencing a subsequent preterm birth. The absolute risks presented in this study can also be used to initiate discussions regarding management plans to minimise risk of preterm birth in subsequent pregnancy.

## Acknowledgments

The authors wish to thank the Healthy Pregnancies WA Consumer Reference Group and thank the Data Linkage Unit of the WA Department of Health for data provision.

## Author Contributions

**Conceptualization:** Elizabeth Pereira, Gavin Pereira.

**Formal analysis:** Gavin Pereira.

**Investigation:** Gizachew Tessema, Mika Gissler, Annette K. Regan.

**Methodology:** Elizabeth Pereira, Gavin Pereira.

**Supervision:** Annette K. Regan.

**Writing – original draft:** Elizabeth Pereira.

**Writing – review & editing:** Elizabeth Pereira, Gizachew Tessema, Mika Gissler, Annette K. Regan, Gavin Pereira.

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
