## [Decision Letter · Decision Letter 0]

16 Dec 2020

PONE-D-20-29280

Will it happen to me again? Re-evaluation of gestational age as a predictor for subsequent preterm birth

PLOS ONE

Dear Dr. Pereira,

Thank you for submitting your manuscript to PLOS ONE. After careful consideration, we feel that it has merit but does not fully meet PLOS ONE’s publication criteria as it currently stands. Therefore, we invite you to submit a revised version of the manuscript that addresses the points raised during the review process.

We look forward to receiving your revised manuscript.

Kind regards,

Angela Lupattelli, PhD

Academic Editor

PLOS ONE

Reviewers' comments:

**Comments to the Author**

1. Is the manuscript technically sound, and do the data support the conclusions?

Reviewer #1: No

Reviewer #2: Yes

2. Has the statistical analysis been performed appropriately and rigorously? 

Reviewer #1: No

Reviewer #2: Yes

3. Have the authors made all data underlying the findings in their manuscript fully available?

Reviewer #1: No

Reviewer #2: Yes

4. Is the manuscript presented in an intelligible fashion and written in standard English?

Reviewer #1: Yes

Reviewer #2: Yes

5. Review Comments to the Author

Reviewer #1: Plos one review gestational age as a prediction of subsequent pretermbirth.

This was a huge study and compares the risk of a second pregnancy delivering preterm if the first pregnancy was a preterm delivery in western australia. Their conclusions are that preterm birth of the first child is a poor predictor for preterm birth of the second child, however I’m not certain how that conclusion was made, as their data supports the opposite conclusion.

7.1% risk of preterm birth on first pregnancy

which increased to

9.19% to 11.64% risk of second child before 32 weeks if first child before 30 weeks reduced to 7.13% if the first child after 30 weeks

which increased to

13-85% to 16.74% risk of second child before 34 weeks if gestational age of first between 22 and 30 weeks reduced to 12.32% if the first child more than 30 weeks.

which increased to

10.9-34.56% risk of a second child before 37 weeks in the first child less that 37 weeks

If you have a first child born preterm then there is an almost 35% risk of a second child being born preterm versus 7.1%. I’m not certain why that increase is being downplayed.

What was the risk of a second child being born preterm if the first child was born after 37% weeks, I cannot find this analysis and this is what all the numbers should be compared to and not to the preterm birth risk of a first born.

Was the interval between first and second pregnancies accounted for?

In the section, what this study adds, the statement that the median age of the second child is at least 38 week including births for mothers when their first child was extremely preterm is misleading and again downplaying the increased risks. Of course the median age in 38 weeks if even 35% of the babies were born preterm. What was the median age of the babies born preterm after the first birth was preterm? Did that increase or decrease?

Reviewer #2: Summary (taken from the abstract).

The authors have analyzed a large retrospective cohort of women who gave birth to their first two children in Western Australia (N=255,151 mothers). They calculated for each week of final gestational age of the first birth, unadjusted relative risks (RR) and absolute risks (AR) of

The relative risks of second birth before 28-, 32-, and 34-weeks’ gestation were all approximately

twenty times higher for mothers whose first birth had a gestational age of 22 to 30 weeks compared to those whose first birth was at 40 weeks’ gestation. The absolute risks of second birth before 28-, 32-, and 34-weeks’ gestation for these mothers were all less than 16.74%. The absolute risk of second birth before 37 weeks was highest at 32.11% (95% CI: 30.27, 34.02) for mothers whose first birth was 22 to 30 weeks’ gestation. For all gestational ages of the first child, the lowest quartile and median gestational age of the second birth were at least 36 weeks and at least 38 weeks, respectively. Sensitivity and positive predictive values were all below 35%. The authors conclude that early gestational age is a strong risk factor but a poor predictor of subsequent preterm birth.

General impression:

This is the first large study assessing the confidence intervals of both the relative risk and the absolute risk of recurrent preterm birth after a first preterm singleton delivery. Strengths of the study include large population-based study with best obstetrical estimate of gestational age. Methods are well explained, data are clear, and the manuscript is easy to follow.

Limitations include lack of analysis of other factors for subsequent preterm pregnancy beyond gestational age, long duration of data collection and lack of testing summary statistics in another population.

Major comments:

1. The authors should state other factors that may affect the risk of recurrent prematurity and list relevant literature. Previous authors (Yang, Obstet Gynecol 2017; Kalengo, PLOS One 2020; Phillips, BMJ Open 2017, and others listed in the manuscript) have shown that other factors may affect the risk of recurrent preterm delivery, e.g., preeclampsia, maternal age <18 years, preterm labor vs. preterm premature rupture of membranes. If this information is not available, it should be listed as a limitation.

2. The study lasted 35 years. It is likely that practice of pregnancy dating and prevention of preterm birth has changed during this time. Concerns were raised about validity of gestational age in the 1986 report listed on ww2.health.wa.gov.au/article

3. Lack of testing sensitivity, specificity and positive predictive value in another population is a major limitation. Values provided only apply to the population from which these data were calculated; they may not apply to a prospective current population.

4. The discussion lacks a paragraph on limitations.

Minor comments:

1. In the abstract, results, the second sentence is not accurate. It would be more exact to state the following: “... had upper confidence interval limit equal or less than 16.74%”.

2. What is known on this subject: Several studies have reported the absolute risk of recurrence of prematurity after a first preterm delivery. It would be correct to state that the confidence intervals of the absolute risk of recurrence of prematurity after a first pregnancy have not been reported.

6. PLOS authors have the option to publish the peer review history of their article (what does this mean?). If published, this will include your full peer review and any attached files.

Reviewer #1: No

Reviewer #2: No

---

## [Author Response · Author response to Decision Letter 0]

4 Jan 2021

Response to Editor (E)

We thank Dr Lupattelli for the time and effort to facilitate the review our manuscript. We have responded to each of the Editor’s comments below. Where changes have been made to the manuscript we refer to line numbers corresponding to the Tracked Changes version of the resubmission. 

E1. Please ensure that your manuscript meets PLOS ONE's style requirements, including those for file naming. The PLOS ONE style templates can be found at

Response to E1. Thank you for pointing us to the style templates. We have revised the submission to meet the style requirements. File names are now “Response to Reviewers.docx” (this document), “Manuscript” (clean copy), “Revised Manuscript with Tracked Changes” (track change copy) and “Fig1.tif” (new file for the figure). The title page, headings, line spacing, line and page numbering and reference style have also been amended to meet the style requirements.

E2. We note that you have indicated that data from this study are available upon request. PLOS only allows data to be available upon request if there are legal or ethical restrictions on sharing data publicly. For information on unacceptable data access restrictions, please see http://journals.plos.org/plosone/s/data-availability#loc-unacceptable-data-access-restrictions.

Response to E2. As requested, we have included this information in the cover letter on the previous page. Although we are proponents for data sharing, we cannot upload these data due to legal restrictions in the form of a data use agreement with the WA Department of Health for the reasons described in the Data Availability statement in our cover letter on the previous page. These factors are reasonable and are not listed in PLOS ONE’s Unacceptable Data Access Restrictions.

E3. Your ethics statement should only appear in the Methods section of your manuscript. If your ethics statement is written in any section besides the Methods, please move it to the Methods section and delete it from any other section. Please ensure that your ethics statement is included in your manuscript, as the ethics statement entered into the online submission form will not be published alongside your manuscript.

Response to E3. As requested, we have moved this statement to the Variables and Data Sources sub-section of the Methods section in the revised version of the manuscript (lines 93-94).

 

Response to Reviewer 1 (R1)

We thank Reviewer 1 for the time and effort to review our manuscript. We have responded to each comment below. Where changes have been made to the manuscript we refer to line numbers corresponding to the Tracked Changes version of the resubmission. 

R1.1. Is the manuscript technically sound, and do the data support the conclusions? Reviewer #1: No (Reviewer #2: Yes)

Response to R1.1. Our manuscript is technically sound. The reviewer raised one technical comment (R1.6), to which we have now responded. The other comments made by the reviewer are about interpretation of the results not our analytical/technical approach. 

The data do support the conclusions. Abridged summary:

What we observed What we concluded

We reported high relative risks lines 28-30 Gestational age is a strong risk factor lines 37-38

Absolute risks, positive predictive values and sensitivities for clinically significant preterm birth were all below 35% lines 30-35 Early gestational age is a poor predictor lines 38-40

The reviewer has confused risk factors (indicated by relative risks) with predictors (indicated by absolute risks and prediction metrics). Please see our response to comment R1.4 (last paragraph). To avoid confusion we have added a paragraph to the beginning of the Statistical Analysis sub-section to make clear the distinction between absolute and relative risk in the revised version of the manuscript (lines 98-106). Please also see our responses to the specific comments below.

R1.2. Has the statistical analysis been performed appropriately and rigorously? Reviewer #1: No (Reviewer #2: Yes)

Response to R1.2. As can be seen from the reviewer’s specific comments (reproduced in this response document verbatim) the reviewer has not made any specific comments regarding the statistical analyses. To our knowledge, our manuscript conforms with PLoS ONE’s Statistical Reporting Guidelines.

R1.3. This was a huge study and compares the risk of a second pregnancy delivering preterm if the first pregnancy was a preterm delivery in western australia. Their conclusions are that preterm birth of the first child is a poor predictor for preterm birth of the second child, however I’m not certain how that conclusion was made, as their data supports the opposite conclusion.

Response to R1.3. The reviewer implies that the gestational age of the first child is a strong risk factor for preterm birth of the second child. We agree. This is stated in our results and conclusion (See our response to R1.1). For example, the risk of second birth at <28 weeks if the first child was 22-30 weeks was 19 times higher than the risk of second birth <28 weeks if the first child was 40 weeks (Table 2). This is a very large relative risk. Does this mean that these mothers are highly likely to deliver their second child at <28 weeks? No. Fewer than one in twenty of these mothers will deliver their second child at <28 weeks (Absolute Risk 4.63%, Table 2). Why? Because 19 times a very small number is still a very small number i.e. only 0.27% of mothers whose first child was 40 weeks delivered their second child <28 weeks (Table 2). 

Similarly, the risk of second birth at <37 weeks if the first child was 22-30 weeks was 10.05 times higher than the risk of second birth <37 weeks if the first child was 40 weeks (Table 2). This is a large relative risk. Does this mean that these mothers are highly likely to deliver their second child at <37 weeks? Again, the answer is ‘No’. Fewer than one in three of these mothers will deliver their second child at <37 weeks (Absolute Risk 32.11%, Table 2). 

We reported other important metrics that support the conclusion that preterm birth of the first child is a poor predictor for preterm birth of the second child (despite being a strong risk factor). Only 2.66% and 27.37% of mothers’ second births that are born <37 weeks had a first birth <28 weeks and <37 weeks, respectively (Sensitivity, Table 3). High proportions of mothers who delivered the first child early delivered at term for their second birth: 68.46% and 77.42% of mothers whose first child was born <28 weeks and <37 weeks, respectively, went on to deliver a child that was not born <37 weeks (1 minus Positive Predictive Value, Table 3).

Therefore, (i) our results suggest that preterm birth of the first child is a strong risk factor for preterm birth of the second child based on the observed relative risks (Table 2); and, (ii) our results suggest that preterm birth of the first child is a poor predictor for preterm birth of the second child based on the observed absolute risks (Table 2) and metrics for predictive performance (Table 3).

To avoid confusion we have added a paragraph to the beginning of the Statistical Analysis sub-section to make clear the distinction between absolute and relative risk in the revised version of the manuscript (lines 98-106). We believe this additional text will help avoid the misinterpretation that “high” relative risk implies strong predictive performance. Further description of the relevance of absolute risks is described at lines 222-226 of the Discussion section.

R1.4. 7.1% risk of preterm birth on first pregnancy which increased to 9.19% to 11.64% risk of second child before 32 weeks if first child before 30 weeks reduced to 7.13% if the first child after 30 weeks which increased to 13-85% to 16.74% risk of second child before 34 weeks if gestational age of first between 22 and 30 weeks reduced to 12.32% if the first child more than 30 weeks. which increased to 10.9-34.56% risk of a second child before 37 weeks in the first child less that 37 weeks. If you have a first child born preterm then there is an almost 35% risk of a second child being born preterm versus 7.1%. I’m not certain why that increase is being downplayed.

Response to R1.4. We have reproduced the results presented in the original submission because the omission of punctuation makes this reviewer comment difficult to interpret and the typographical errors overstates risk e.g., “which increased [risk] to 13-85%”. These are the results that we presented in the “Absolute risk of preterm birth of the second child by gestational age of the first child” sub-section of the Results in the original submission:

“The absolute risk of second child birth before 32 weeks’ gestation ranged from 9.19% to 11.64% if the gestational age of the first child was between 22 and 30 weeks, and less than 7.13% if the gestational age of the first child was more than 30 weeks. The interval estimates of absolute risk of second child birth before 34 weeks’ gestation ranged from 13.85% to 16.74% if the gestational age of the first child was between 22 and 30 weeks, and was less than 12.32% if the gestational age of the first child was more than 30 weeks. In contrast, the absolute risk of second child birth before 37 weeks’ gestation was elevated for all gestational ages of the first child less than 37 weeks, ranging from 10.90% to 34.56%.”

Again, we agree with the reviewer. The “increase” is notable. This “increase” is reported in our study via the “relative risk”, which we have interpreted in the “Relative risk of preterm birth of the second child by gestational age of the first child” sub-section of the Results. The reviewer is interpreting the change in absolute risk estimates, which by definition is a relative risk. In contrast, the results presented in the “Absolute risk of preterm birth of the second child by gestational age of the first child” sub-section of the Results should be interpreted in terms of their absolute magnitude. For every comparison investigated, predictions of second child preterm birth based on first child preterm birth were all < 35%, which is much lower than the probability from the flip of a coin. Of course, other metrics such as sensitivity and positive predictive value, are relevant but these were also very low (see the paragraph in our response to R1.3). 

R1.5. What was the risk of a second child being born preterm if the first child was born after 37% weeks, I cannot find this analysis and this is what all the numbers should be compared to and not to the preterm birth risk of a first born.

Response to R1.5. We agree with the reviewer’s recommendation of the reference category, and stress that we did this in the original submission. The results are reported in Table 2. The relative risk estimates use full term (40 weeks) as the reference category. We have reported relative risks of preterm birth for the second child when the gestational age of the first birth was 22-30 weeks, 31-32 weeks, each week from 33 to 41 weeks (except for 40 weeks, because it was the reference category), and 42-44 weeks. 

We are unsure if the reviewer is aware how to interpret “relative risks” and have provided the following example for their benefit (italicized text). In this example we interpret the relative risk of 8.98, reported in Table 2. This relative risk of 8.98 is when the first child was born at 33 weeks of gestation and the second child is born <37 weeks. The interpretation is that the probability of second child birth <37 weeks when the first child is born at 33 weeks is 8.98 times the probability of second child birth <37 weeks when the first child is born at 40 weeks. 

R1.6. Was the interval between first and second pregnancies accounted for?

Response to R1.6. This is a reasonable question given the dominance of etiological studies in the published literature. We agree with the reviewer that adjustment for potential confounders, including but not limited to interpregnancy intervals, are likely to attenuate the relative risks. However, our study was not an etiological study. The aim of our study was to evaluate gestational age of the first child as a predictor for preterm birth of the second child (Abstract, Background, and Methods). Adjustment is not relevant and the rationale for this is described in the manuscript (lines 121 – 124 of the Statistical Analysis sub-section). We reported relative risks alongside absolute risks and prediction metrics for completeness. 

R1.7. In the section, what this study adds, the statement that the median age of the second child is at least 38 week including births for mothers when their first child was extremely preterm is misleading and again downplaying the increased risks. Of course the median age in 38 weeks if even 35% of the babies were born preterm. What was the median age of the babies born preterm after the first birth was preterm? Did that increase or decrease?

Response to R1.7. We have removed this section to conform to the journal’s requirements. We made a small amendment to make the title less subjective. We did not intend for the statement to which the reviewer refers to be misleading and do not believe our results are misleading because the median gestational ages of the second birth by gestational age of the first birth were reported in the original and revised submissions for all gestational ages of the first birth (Table 2). We also did not limit our results to the median, and believe we were thorough and highly transparent in reporting multiple quantiles (5th, 10th, 15th, 25th, 50th, 75th, 90th centiles). 

To answer the reviewer’s specific question, the median gestational age of the second birth was 38 - 40 weeks and therefore changed very little. 

However, as we have previously stated (see response to R1.4) the “increase” or “decrease” in gestational age and chance of preterm birth across the range of gestational ages of the first birth is a relative measure of risk. Again, we agree with the reviewer that relative risks of preterm birth increase with decreasing gestational length of the first child. This has been reported by others as well. However, our results show that gestational length of the first child is a poor predictor of preterm birth of the second child (see Absolute Risks in Table 2 and prediction metrics in Table 3) despite it being a strong risk factor (see Relative Risks in Table 2). 

Response to Reviewer 2 (R2)

We thank Reviewer 2 for the time and effort to review our manuscript. We have responded to each comment below. Where changes have been made to the manuscript we refer to line numbers corresponding to the Tracked Changes version of the resubmission. 

R2.1 General impression: This is the first large study assessing the confidence intervals of both the relative risk and the absolute risk of recurrent preterm birth after a first preterm singleton delivery. Strengths of the study include large population-based study with best obstetrical estimate of gestational age. Methods are well explained, data are clear, and the manuscript is easy to follow. Limitations include lack of analysis of other factors for subsequent preterm pregnancy beyond gestational age, long duration of data collection and lack of testing summary statistics in another population.

Response to R2.1. We appreciate the reviewer’s positive comments and would like to raise attention to another strength of the study, namely that preterm birth has been long regarded as a risk factor for subsequent preterm birth but until now has never been evaluated as a predictor. We have addressed the limitations raised by the reviewer in the specific comments below. 

R2.2. The authors should state other factors that may affect the risk of recurrent prematurity and list relevant literature. Previous authors (Yang, Obstet Gynecol 2017; Kalengo, PLOS One 2020; Phillips, BMJ Open 2017, and others listed in the manuscript) have shown that other factors may affect the risk of recurrent preterm delivery, e.g., preeclampsia, maternal age <18 years, preterm labor vs. preterm premature rupture of membranes. If this information is not available, it should be listed as a limitation.

Response to R2.2. We agree with the reviewer that there are multiple risk factors for “recurrent prematurity”, including those mentioned as well as genetic predisposition. We intentionally did not adjust for risk factors. We intentionally allowed the relative risks to reflect both direct effects of first child gestational age as well as indirect effects via other associated risk factors such as those mentioned by reviewer. If we included other risk factors, such as those mentioned by the reviewer, the model may improve its predictive performance but the predictive performance of first child gestational age (which is the primary subject of this study) would decrease. Our relative risk estimates therefore reflect the “total effect” of first child gestational age on risk of preterm birth of the second child. The aim of our study was to evaluate gestational age of the first child as a predictor for preterm birth of the second child (Abstract, Background, and Methods) and therefore adjusting out the effects of other risk factors is not relevant (see lines 121 – 124 of the Statistical Analysis sub-section for further rationale).

R2.3. The study lasted 35 years. It is likely that practice of pregnancy dating and prevention of preterm birth has changed during this time. Concerns were raised about validity of gestational age in the 1986 report listed on ww2.health.wa.gov.au/article

Response to R2.3. We acknowledge the reviewer’s concern and broadly agree with the reviewer. The long duration of data collection is both a strength and a limitation. The long duration allows us to observe larger numbers of births at early gestations (closer to viability) than if the duration of the study was shorter. However, a longer study period is accompanied by temporal changes. The validity of gestational age was mentioned in the 1986 validation study ( https://ww2.health.wa.gov.au/-/media/Files/Corporate/general-documents/Data-collection/PDF/Midwives_Validation_Study_1986.pdf). Specifically, the 1986 validation study states that the estimated gestation on the midwives notification was the same as that stated in the medical record for 79.4% of cases. The 1986 validation study did not report a measure of central tendency such as the mean absolute difference between the two dates or any useful measure of the degree of the difference. The most recent validation study was conducted in 2007 (https://ww2.health.wa.gov.au/-/media/Files/Corporate/general-documents/Data-collection/PDF/Midwives_Validation_Study_2007.pdf, Reference 10). The 2007 validation study reported that estimated gestation was correct for 468 of the 525 records (89%). More importantly, the 2007 study indicated that of the 57 records for which estimated gestation was incorrect, 50 of them were due to a rounding error whereby the midwives notification was rounded to the nearest whole week of gestation instead of rounding down to the completed week of gestation. If we are willing to accept such a small error, this would mean that only 98.7% of the midwives notifications had the correct gestational age.

Gestational age is obtained as the clinical best estimate from the dating ultrasound or a derivation based on birth date and date last menstrual period (LMP), with the latter more common earlier in the cohort period. The 1986 validation study states that birth dates were accurate for 99.7% of the records in the validation study (p13, and Table 2 on p14) and last menstrual period dates were correctly transcribed for 97.1% of the records (on p19). The transcription of these data items is not a limitation. The main limitation is due to the loss of accuracy when using gestational age estimated from LMP compared to gestational ultrasound dating. The consequence of this limitation is likely to have resulted in wider confidence intervals (loss of precision), which we have now described as a limitation in the Discussion. 

We have also described secular changes in antenatal care as a limitation because this could have resulted in prevention of preterm birth. However, we note that the change in antenatal care between each mothers first birth and second birth is likely to be minimal because this interval is of the order of years not decades. Similarly, we mentioned the potential for changes in behavioural risk factors during the study period. Again, we anticipate that these limitations are likely to have resulted in wider confidence intervals (loss of precision), but this would not have changed our conclusion because estimates in Table 3 were already precise for sensitivity and positive predictive values.

The additional description on limitations is now described in the Limitations paragraph, at lines 241 - 246 of the revised manuscript, which is the last paragraph of the Discussion section.

R2.4. Lack of testing sensitivity, specificity and positive predictive value in another population is a major limitation. Values provided only apply to the population from which these data were calculated; they may not apply to a prospective current population.

Response to R2.4. We thank the reviewer for raising this point. In response, we have now mentioned the fact that our results need to be replicated/tested in an independent study population at lines 246 - 250 in the Limitations paragraph, which is the last paragraph of the Discussion section. We have also referred to this limitation as the main limitation of our study. Our starting study population included almost one million births over more than three decades for the whole state. We do agree that this is a limitation but do not believe it is a major limitation because validation at this scale – replication of a study of this size in an independent population (which is needed to observe enough births at early gestations) - would be a significant undertaking, undertaken in a new study and separately published. 

R2.5. The discussion lacks a paragraph on limitations.

Response to R2.5. Limitations were mentioned in the second last paragraph of the original submission, although we acknowledge that it was sparse. We have included further description of limitations at lines 241 - 250 in the Limitations paragraph, which is the last paragraph of the Discussion section in the revised version of the manuscript. 

R2.6. In the abstract, results, the second sentence is not accurate. It would be more exact to state the following: “... had upper confidence interval limit equal or less than 16.74%”.

Response to R2.6. We appreciate the suggestion and have made the change recommended by the reviewer at line 31 of the Results section.

R2.7. What is known on this subject: Several studies have reported the absolute risk of recurrence of prematurity after a first preterm delivery. It would be correct to state that the confidence intervals of the absolute risk of recurrence of prematurity after a first pregnancy have not been reported.

Response to R2.7. We have removed this section to conform to the journal’s requirements but agree with the reviewer’s comment nonetheless.

---

## [Editor Report · Decision Letter 1]

11 Jan 2021

Re-evaluation of gestational age as a predictor for subsequent preterm birth

PONE-D-20-29280R1

Dear Dr. Pereira,

We’re pleased to inform you that your manuscript has been judged scientifically suitable for publication and will be formally accepted for publication once it meets all outstanding technical requirements.

Kind regards,

Angela Lupattelli, PhD

Academic Editor

PLOS ONE

---

## [Editor Report · Acceptance letter]

13 Jan 2021

PONE-D-20-29280R1 

Re-evaluation of gestational age as a predictor for subsequent preterm birth 

Dear Dr. Pereira:

I'm pleased to inform you that your manuscript has been deemed suitable for publication in PLOS ONE. Congratulations! Your manuscript is now with our production department. 

Kind regards, 

on behalf of

Dr. Angela Lupattelli 

Academic Editor

PLOS ONE